# NEURAL NETWORKS WITH TRAINABLE MATRIX ACTIVATION FUNCTIONS

## ABSTRACT

The training process of neural networks usually optimizes weights and bias parameters of linear transformations, while nonlinear activation functions are pre-specified and fixed. This work develops a systematic approach to constructing matrix activation functions whose entries are generalized from ReLU. The activation is based on matrix-vector multiplications using only scalar multiplications and comparisons. The proposed activation functions depend on parameters that are trained along with the weights and bias vectors. Neural networks based on this approach are simple and efficient and are shown to be robust in numerical experiments.

## 1 INTRODUCTION

In recent decades, deep neural networks (DNNs) have achieved significant successes in many fields such as computer vision and natural language processing (Voulodimos et al., 2018), (Otter et al., 2018). The DNN surrogate model is constructed using recursive composition of linear transformations and nonlinear activation functions. To achieve good performance, it is essential to choose activation functions suitable for specific applications. In practice, Rectified Linear Unit (ReLU) is one of the most popular activation functions for its simplicity and efficiency. A drawback of ReLU is the presence of vanishing gradient in the training process, known as dying ReLU problem (Lu et al., 2019). Several relatively new activation approaches are proposed to overcome this problem, e.g., the simple Leaky ReLU, and Piecewise Linear Unit (PLU) (Nicolae, 2018), Softplus (Glorot et al., 2011), Exponential Linear Unit (ELU) (Clevert et al., 2016), Scaled Exponential Linear Unit (SELU) (Klambauer et al., 2017), Gaussian Error Linear Unit (GELU) (Hendrycks & Gimpel, 2016).

Although the aforementioned activation functions are shown to be competitive in benchmark tests, they are still fixed nonlinear functions. In a DNN structure, it is often hard to determine a priori the optimal activation function for a specific application. In this paper, we shall generalize ReLU and introduce arbitrary trainable matrix activation functions. The effectiveness of the proposed approach is validated using function approximation examples and well-known benchmark datasets such as `CIFAR-10` and `CIFAR-100`. There are a few classical works on adaptively tuning of parameters in the training process, e.g., the parametric ReLU (He et al., 2015b). However, our adaptive matrix activation functions are shown to be competitive and more robust in those experiments.

### 1.1 PRELIMINARIES

We consider the general learning process based on a given training set $\{(x_n, f_n)\}_{n=1}^N$, where the inputs $\{x_n\}_{n=1}^N \subset \mathbb{R}^d$ and outputs $\{f_n\}_{n=1}^N \subset \mathbb{R}^J$ are implicitly related via an unknown target function $f : \mathbb{R}^d \to \mathbb{R}^J$ with $f_n = f(x_n)$. The ReLU activation function is a piecewise linear function given by

$$\sigma(t) = \max\{t, 0\}, \quad \text{for} \quad t \in \mathbb{R}. \tag{1}$$

In the literature $\sigma$ is acting component-wise on an input vector. In a DNN, let $L$ be the number of layers and $n_\ell$ denote the number of neurons at the $\ell$-th layer for $0 \le \ell \le L$ with $n_0 = d$ and $n_L = J$. Let $\mathcal{W} = (W_1, W_2, \ldots, W_L) \in \prod_{\ell=1}^L \mathbb{R}^{n_\ell \times n_{\ell-1}}$ denote the tuple of admissible weight matrices and $\mathcal{B} = (b_1, b_2, \ldots, b_L) \in \prod_{\ell=1}^L \mathbb{R}^{n_\ell}$ the tuple of admissible bias vectors. The ReLU

DNN approximation to $f$ at the $L$-th layer is recursively defined as

$$\eta_L = \eta_{L,\mathcal{W},\mathcal{B}} := h_{W_L,b_L} \circ \sigma \circ \cdots \circ h_{W_2,b_2} \circ \sigma \circ h_{W_1,b_1}, \tag{2}$$

where $h_{\mathcal{W}_\ell,b_\ell}(x) = W_\ell x + b_\ell$ is an affine linear transformation at the $\ell$-th layer, and $\circ$ denotes the composition of functions. The traditional training process for such a DNN is to find optimal $\mathcal{W}_* \in \prod_{\ell=1}^{L} \mathbb{R}^{n_\ell \times n_{\ell-1}}$, $\mathcal{B}_* \in \prod_{\ell=1}^{L} \mathbb{R}^{n_\ell}$, (and thus optimal $\eta_{L,\mathcal{W}_*,\mathcal{B}_*}$) such that

$$(\mathcal{W}_*, \mathcal{B}_*) = \arg\min_{\mathcal{W},\mathcal{B}} \frac{1}{N} \sum_{n=1}^{N} |f_n - \eta_{L,\mathcal{W},\mathcal{B}}(x_n)|^2. \tag{3}$$

In other words, $\eta_{L,\mathcal{W}_*,\mathcal{B}_*}$ best fits the data with respect to the $\ell^2$ norm within the function class $\{\eta_{L,\mathcal{W},\mathcal{B}}\}$. In practice, the sum of squares norm used in minimization could be replaced with other convenient norms.

## 2 TRAINABLE MATRIX ACTIVATION FUNCTION

Having a closer look at ReLU $\sigma$, we observe that the activation $\sigma \circ \eta_\ell(x) = \sigma(\eta_\ell(x))$ could be realized as a matrix-vector multiplication $\sigma \circ \eta_\ell(x) = D_\ell(\eta_\ell(x))\eta_\ell(x)$ or equivalently $\sigma \circ \eta_\ell = (D_\ell \circ \eta_\ell)\eta_\ell$, where $\eta_\ell$ is *column* vector-valued in $\mathbb{R}^{n_\ell}$ and $D_\ell$ is a *diagonal* matrix-valued function mapping from $\mathbb{R}^{n_\ell}$ to $\mathbb{R}^{n_\ell \times n_\ell}$ with entries from the discrete set $\{0, 1\}$. This is a simple but quite useful observation. There is no reason to restrict on $\{0, 1\}$ and we thus look for a larger set of values over which the diagonal entries of $D_\ell$ are running or sampled. With slight abuse of notation, our new DNN approximation to $f$ is calculated using the following recurrence relation

$$\eta_1 = h_{W_1,b_1}, \quad \eta_{\ell+1} = h_{W_{\ell+1},b_{\ell+1}} \circ \big[(D_\ell \circ \eta_\ell)\eta_\ell\big], \quad \ell = 1, \ldots, L-1. \tag{4}$$

Here each $D_\ell$ is diagonal and is of the form

$$D_\ell(y) = \mathrm{diag}(\alpha_\ell(y_1), \alpha_\ell(y_2), \ldots, \alpha_\ell(y_{n_\ell})), \quad y \in \mathbb{R}^{n_\ell}, \tag{5}$$

where $\alpha_\ell$ is a nonlinear function to be determined. Since piecewise constant functions can approximate a continuous function within arbitrarily high accuracy, we choose $\alpha_\ell$ to be the following step function

$$\alpha_\ell(s) = \begin{cases} t_{\ell,0}, & s \in (-\infty, s_1], \\ t_{\ell,1}, & s \in (s_1, s_2], \\ \quad \vdots & \\ t_{\ell,m-1}, & s \in (s_{m-1}, s_m], \\ t_{\ell,m}, & s \in (s_m, \infty), \end{cases} \tag{6}$$

where $m$ is a positive integer and $\{t_{\ell,j}\}_{j=0}^{m}$ and $\{s_j\}_{j=1}^{m}$ are constants. For the time being, let $D_\ell(\eta_\ell(x))\eta_\ell(x) = \sigma_\ell(\eta_\ell(x))$ with $\sigma_\ell$ being a scalar-valued function applied to $\eta_\ell(x)$ component-wise. In the following, we list several $\sigma_\ell$s corresponding to special $\alpha_\ell$ and visualize them in Figure 1.

$$m = 1, s_1 = 0, t_{\ell,0} = 0, t_{\ell,1} = 1 \implies \sigma_\ell \text{ is ReLU},$$
$$m = 1, s_1 = 0, t_{\ell,0} > 0, t_{\ell,1} = 1 \implies \sigma_\ell \text{ is Leaky ReLU},$$
$$m = 2, s_1 = 0, s_2 = 1, t_{\ell,0} = t_{\ell,2} = 0, t_{\ell,1} = 1 \implies \sigma_\ell \text{ is discontinuous}.$$

In our case, we will choose the grid points $\{s_j\}_{j=1}^{m}$ a priori and train the step values $\cup_{\ell=1}^{L}\{t_{\ell,j}\}_{j=0}^{m}$ of $\alpha_\ell$. In such a way, the resulting DNN may use different activation functions for different layers, and these activation functions are naturally adapted to the target function $f$ and the target dataset. Since ReLU and Leaky ReLU are included by our DNN as special cases, the proposed DNN is clearly not worse than the traditional ones in practice. In the following, we call the neural network in equation 4, with the activation approach given in equation 5 and equation 6, a DNN based on the "Trainable Matrix Activation Function (TMAF)".

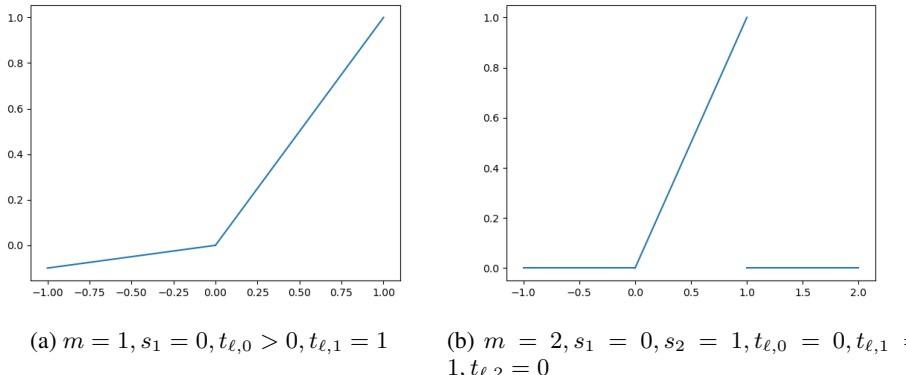

(a) $m = 1, s_1 = 0, t_{\ell,0} > 0, t_{\ell,1} = 1$

(b) $m = 2, s_1 = 0, s_2 = 1, t_{\ell,0} = 0, t_{\ell,1} = 1, t_{\ell,2} = 0$

Figure 1: Graphs of $\sigma_\ell$.

REMARK

*The activation functions used in TMAF neural network are not piecewise constants. Instead, TMAF activation is realized using matrix-vector multiplication, where entries of those matrices are trainable piecewise constants. There exist several trainable activation functions in the literature, e.g., Parametric ReLU (He et al., 2015b), Padé Activation Unit (PAU) (Molina et al., 2020), Piecewise Linear Unit (PWLU) (Zhou et al., 2021), Swish (Ramachandran et al., 2017), adaptive rational functions (Boullé et al., 2020), adaptive splines (Bohra et al., 2020), adaptive activation functions for physics informed neural networks (ada, 2020). We point out that TMAF is different from the aforementioned activation approaches. One might think TMAF is similar to PWLU because they are both defined piecewise. However, by construction TMAF includes discontinuous functions as special cases (see Figure 1) while PWLU always results in a* continuous *piecewise linear function with trainable slopes.*

Starting from the diagonal activation $D_\ell$, we can go one step further to construct more general activation matrices. First we note that $D_\ell$ could be viewed as a nonlinear operator $T_\ell : [C(\mathbb{R}^d)]^{n_\ell} \to [C(\mathbb{R}^d)]^{n_\ell}$, where

$$[T_\ell(g)](x) = D_\ell(g(x))g(x), \quad g = (g_1, \ldots, g_{n_\ell})^\top \in [C(\mathbb{R}^d)]^{n_\ell}, \quad x \in \mathbb{R}^d.$$

There seems to be no special reason, aside from keeping the computational cost as low as possible, to consider only diagonal operators $D$. A more ambitious approach is to consider a trainable nonlinear activation *operator* determined by more general matrices, for example, by a tri-diagonal operator

$$[T_\ell(g)](x) = \begin{pmatrix} \alpha_\ell(g_1(x)) & \beta_\ell(g_2(x)) & 0 & \cdots & 0 \\ \gamma_\ell(g_1(x)) & \alpha_\ell(g_2(x)) & \beta_\ell(g_3(x)) & \cdots & 0 \\ \vdots & \ddots & \ddots & \ddots & \vdots \\ 0 & 0 & \cdots & \alpha_\ell(g_{n_\ell-1}(x)) & \beta_\ell(g_{n_\ell}(x)) \\ 0 & 0 & \cdots & \gamma_\ell(g_{n_\ell-1}(x)) & \alpha_\ell(g_{n_\ell}(x)) \end{pmatrix} g(x), \quad (7)$$

for $x \in \mathbb{R}^d$. The diagonal entry $\alpha_\ell$ is given in equation 6 while the off-diagonal entries $\beta_\ell$, $\gamma_\ell$ are piecewise constant functions defined in a fashion similar to $\alpha_\ell$. Theoretically speaking, even trainable full matrix activation is possible despite of potentially huge training cost. In summary, a DNN based on trainable nonlinear activation operators $\{T_\ell\}_{\ell=1}^L$ reads

$$\eta_1 = h_{W_1,b_1}, \quad \eta_{\ell+1} = h_{W_{\ell+1},b_{\ell+1}} \circ T_\ell(\eta_\ell), \quad \ell = 1, \ldots, L-1. \quad (8)$$

As we indicated above, TMAF can be used with any matrix and this will lead to a great flexibility in training and approximation. In fact, if a sparse matrix $D$ is used, so that the matrix-vector multiplication is proportional to the number of rows in the matrix, then the resulting algorithm will be computationally efficient. We can choose p-diagonal matrices, as well as matrices with prescribed sparsity patterns. Clearly, based on the simple examples we have considered here, one may conclude that such possibilities, when investigated in depth, can increase the robustness and the accuracy of the TMAF-NN.

REMARK

*Our observation also applies to an activation function $\sigma$ other than ReLU. For example, we may rescale $\sigma(x)$ to obtain $\sigma(\omega_{i,\ell}x)$ using a set of constants $\{\omega_{i,\ell}\}_{1 \leq i \leq n_\ell, 1 \leq \ell \leq L}$ varying layer by layer and neuron by neuron. Then $\sigma(\omega_{i,\ell}x)$ are used to form a matrix activation function and a TMAF DNN, where $\{\omega_{i,\ell}\}$ are trained according to given data and are adapted to the target function.*

## 3 NUMERICAL RESULTS

In this section, we demonstrate the feasibility and efficiency of TMAF by comparing it with traditional ReLU-type activation functions. Recall that neurons in the $\ell$-th layer will be activated by the matrix $D_\ell$. In principle, all parameters in equation 6 are allowed to be trained. To ensure practical efficiency, we shall fix interval grid points $\{s_j\}_{j=1}^m$ used in $\alpha_\ell$ and only let function values $\{t_{\ell,j}\}$ in equation 6 be trained in the following. In each experiment, we use the same learning rates, stochastic optimization methods, and number NE of epochs (optimization iterations). In particular, the learning rate is 1e-4 from epoch 1 to $\frac{\text{NE}}{2}$ and 1e-5 is used from epoch $\frac{\text{NE}}{2} + 1$ to NE. The optimization method is ADAM (Kingma & Ba (2015)) based on mini-batches of size 500. Numerical experiments are performed in PyTorch (Paszke et al. (2019)). We provide two sets of numerical examples:

- Function approximations by TMAF networks and ReLU-type networks;
- Classification problems for `MNIST` and `CIFAR` set solved by TMAF and ReLU networks.

For the first class of examples we use the $\ell^2$-loss function as defined in equation 3. For the classification problems we consider the *cross-entropy* that is widely used as a loss function in classification models. The cross entropy is defined using a training set having $p$ images, each with $N$ pixels. Thus, the training dataset is equivalent to the vector set $\{z_j\}_{j=1}^p \subset R^N$ with each $z_j$ being an image. The $j$-th image belongs to a class $c_j \in \{1, \ldots, M\}$. The neural network maps $z_j$ to $x_j \in \mathbb{R}^M$,

$$x_j := \eta_{L,\mathcal{W},\mathcal{B}}(z_j) \in \mathbb{R}^M, \quad z_j \in \mathbb{R}^N, \quad j = 1, \ldots, p.$$

The cross entropy loss function of $\eta_{L,\mathcal{W},\mathcal{B}}$ then is defined by

$$\mathcal{E}(\mathcal{W}, \mathcal{B}) = -\frac{1}{p} \sum_{k=1}^p \log \left( \frac{\exp(x_{c_k,k})}{\sum_{j=1}^M \exp(x_{j,k})} \right).$$

### 3.1 APPROXIMATION OF A SMOOTH FUNCTION

As our first example, we use neural networks to approximate

$$f(x_1, \cdots, x_n) = \sin(\pi x_1 + \cdots + \pi x_n), \quad x_k \in [-2, 2], \quad k = 1, \ldots, n.$$

The training datasets consist of 20000 input-output data pairs where the input data are randomly sampled from the hypercube $[-2, 2]^n$ based on uniform distribution. The neural networks have single or double hidden layers. Each layer (except input and output layers) has 20 neurons. For TMAF $D_\ell$ in equation 5, the function $\alpha_\ell$ uses intervals $(-\infty, -1.4)$, $(-1.4, -0.92]$, $(-0.92, -0.56]$, $(-0.56, -0.26]$, $(-0.26, 0]$, $(0, 0.26]$, $(0.26, 0.56]$, $(0.56, 0.92]$, $(0.92, 1.4]$, $(1.4, \infty)$ such that probability over each of the ten intervals is 0.1 with respect to Gaussian distribution. Moreover, we apply `BatchNorm1d` in PyTorch to the linear output of neural networks in each hidden layer. The approximation results are shown in Table 1 and Figures 2–3, where Para-ReLU stands for the parametric ReLU neural network. It is observed that TMAF is the most accurate activation approach in these examples.

### 3.2 APPROXIMATION OF AN OSCILLATORY FUNCTION

The next example is on approximating the following function having high, medium and low frequency components

$$f(x) = \sin(100\pi x) + \cos(50\pi x) + \sin(\pi x), \tag{9}$$

see Figure 4a for an illustration. In fact, the function in equation 9 is rather difficult to capture by traditional approximation methods as it is highly oscillatory. We consider ReLU, parametric ReLU,

|  | Approximation error | | | |
|---|---|---|---|---|
|  | Single hidden layer | | Two hidden layers | |
| $n$ | 1 | 2 | 5 | 6 |
| ReLU | 0.09 | 0.34 | 0.14 | 0.48 |
| Para ReLU | 0.04 | 0.11 | 0.09 | 0.47 |
| TMAF | 0.01 | 0.05 | 0.02 | 0.13 |

Table 1: Approximation errors for $\sin(\pi x_1 + \cdots + \pi x_n)$ by neural networks

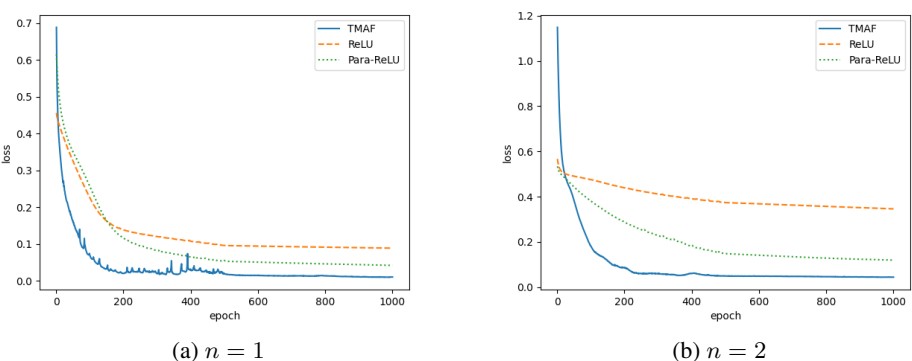

(a) $n = 1$       (b) $n = 2$

Figure 2: Training errors for $\sin(\pi x_1 + \cdots + \pi x_n)$, single hidden layer

and diagonal TMAF neural networks with 3 hidden layers and 400 neurons per layer (except the first and last layers). We also consider the tri-diagonal TMAF (denoted by Tri-diag TMAF, see equation 7) with 3 hidden layers and 300 neurons per layer. The training datasets are 20000 input-output data pairs where the input data are randomly sampled from the interval $[-1, 1]$ based on uniform distribution.

The diagonal TMAF uses $\alpha = \alpha_\ell$ in equation 6 with intervals $(-\infty, -1], (-1+kh, -1+(k+1)h]$, $(1, \infty)$ for $h = 0.02$, $0 \le k \le 99$. The tri-diagonal TMAF is given in equation 7, where $\alpha_\ell$ is the same as the diagonal TMAF, while $\beta_\ell$ is piecewise constant with respect to intervals $(-\infty, -2.01 + \underline{h}], \{(-2.01 + kh, -2.01 + (k+1)h]\}_{k=0}^{99}, (-0.01, \infty)$, and $\gamma_\ell$ is a piecewise constant based on $(-\infty, 0.01], \{(0.01 + kh, 0.01 + (k+1)h]\}_{k=0}^{99}, (2.01, \infty)$, respectively. Numerical results could be found in Figures 4b, 5, 6 and Table 2.

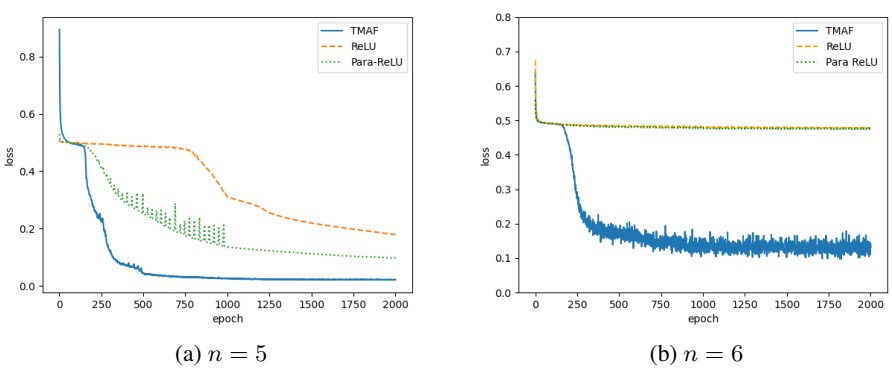

(a) $n = 5$       (b) $n = 6$

Figure 3: Training errors for $\sin(\pi x_1 + \cdots + \pi x_n)$, two hidden layers.

|              | final loss |
|--------------|------------|
| ReLU         | 1.00       |
| Para ReLU    | 1.00       |
| Diag TMAF    | 6.45e-2    |
| Tri-diag TMAF| 5.81e-2    |

Table 2: Error comparison for $\sin(100\pi x) + \cos(50\pi x) + \sin(\pi x)$

For this challenging problem, we note that the diagonal TMAF and tri-diagonal TMAF produce high-quality approximations (see Figures 5 and 4b) while ReLU and parametric ReLU are not able to approximate the highly oscillating function within reasonable accuracy, see Figure 4b and Table 2. Moreover, it is observed from Figure 6 that ReLU and parametric ReLU actually approximate the low frequency part of the target function. To capture the high frequency, ReLU-type neural networks are clearly required to use much more neurons, introducing significantly amount of weight and bias parameters. On the other hand, increasing the number of intervals in TMAF only lead to a few more training parameters.

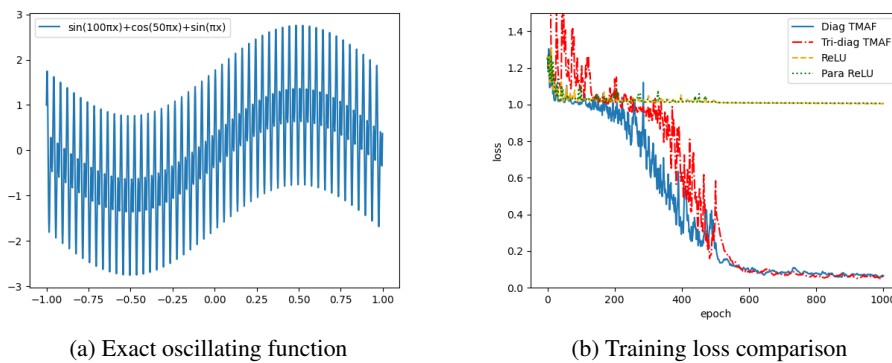

(a) Exact oscillating function        (b) Training loss comparison

Figure 4: Plot of $\sin(100\pi x) + \cos(50\pi x) + \sin(\pi x)$ and training loss comparison

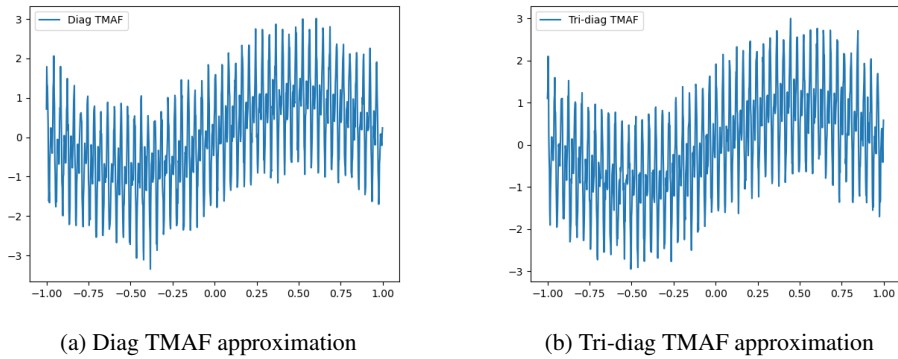

(a) Diag TMAF approximation        (b) Tri-diag TMAF approximation

Figure 5: Approximations to $\sin(100\pi x) + \cos(50\pi x) + \sin(\pi x)$, TMAF-type

## 3.3 CLASSIFICATION OF MNIST AND CIFAR DATASETS

We now test TMAF by classifying images in the `MNIST`, `CIFAR-10` and `CIFAR-100` dataset.

For the `MNIST` set, we implement double layer fully connected networks defined as in equation 2 and equation 4 with 10 neurons per layer (except at the first layer $n_0 = 764$), and we use ReLU or

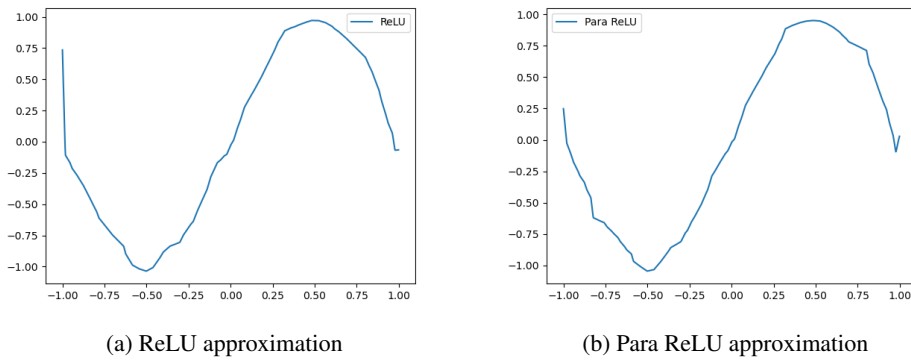

(a) ReLU approximation        (b) Para ReLU approximation

Figure 6: Approximations to $\sin(100\pi x) + \cos(50\pi x) + \sin(\pi x)$, ReLU-type

diagonal TMAF as described in Subsection 3.1. Numerical results are shown in Figures 7a, 7b and Table 3. We observe that performance of the TMAF and the ReLU networks are similar. Such a behavior clearly should be expected for a simple dataset such as `MNIST`.

For the `CIFAR-10` dataset, we use the `ResNet18` network structure provided by (He et al., 2015a). The activation functions are still ReLU and the diagonal TMAF used in Subsection 3.1. Numerical results are presented in Figures 8a, 8b and Table 3. Those parameters given in (Paszke et al., 2019) are already tuned well with respect to ReLU. Nevertheless, TMAF still produces smaller errors in the training process and returns better classification results in the evaluation stage, see Table 3.

For the `CIFAR-100` dataset, we use the `ResNet34` network structure provided by (He et al., 2015a) with the ReLU and TMAF activation functions in Subsection 3.1. Numerical results are presented in Figures 9a and 9b. In the training process, TMAF again outperforms the classical ReLU network.

It is possible to improve the performance of TMAF applied to those benchmark datasets. The key point is to select suitable intervals in $\alpha_\ell$ to optimize the performance. A simple strategy is to let those intervals in equation 6 be varying and adjusted in the training (or the evaluation) process, which will be investigated in our future research.

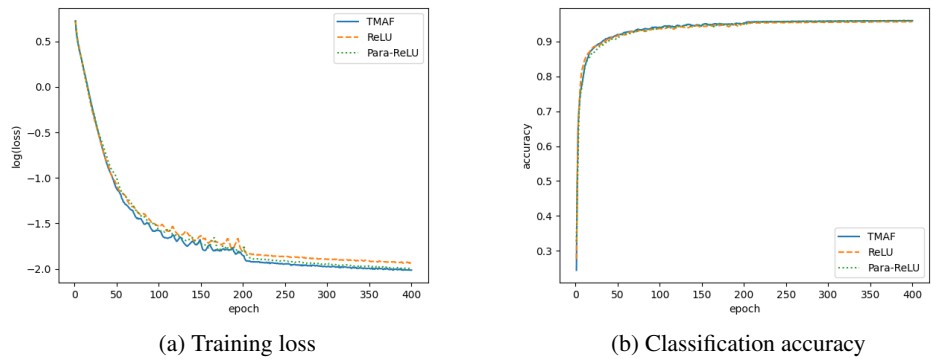

(a) Training loss        (b) Classification accuracy

Figure 7: `MNIST`: Two hidden layers

## REFERENCES

Adaptive activation functions accelerate convergence in deep and physics-informed neural networks. *Journal of Computational Physics*, 404:109136, 2020. ISSN 0021-9991. doi: https://doi.

| Dataset | Evaluation Accuracy | | |
|---------|------|------|-----------|
| | ReLU | TMAF | Para-ReLU |
| MNIST (2 hidden layers) | 91.8% | 92.2% | 91.5% |
| CIFAR-10 (Resnet18) | 77.5% | 80.2% | 78.1% |

Table 3: Evaluation accuracy for CIFAR and MNIST

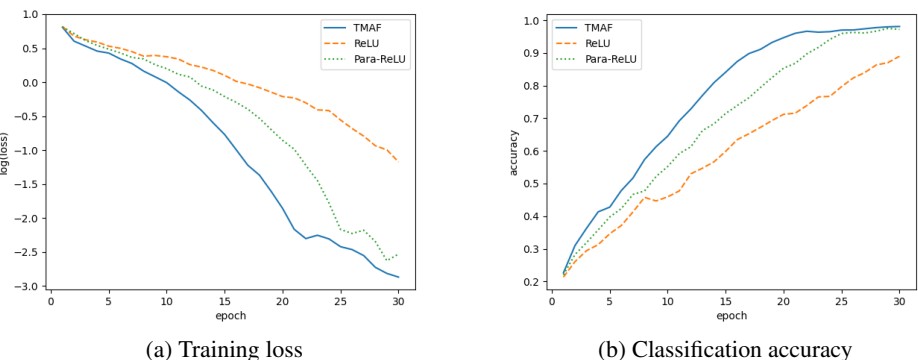

(a) Training loss          (b) Classification accuracy

Figure 8: Comparison among ReLU, Para-ReLU and TMAF for CIFAR-10

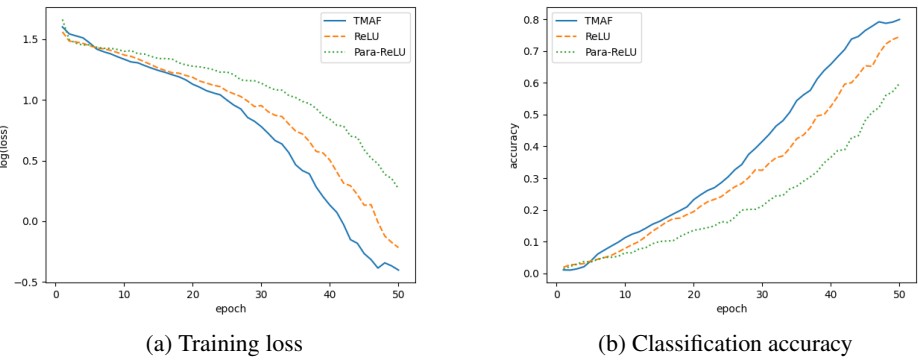

(a) Training loss          (b) Classification accuracy

Figure 9: Comparison among ReLU, Para-ReLU and TMAF for CIFAR-100

org/10.1016/j.jcp.2019.109136. URL https://www.sciencedirect.com/science/article/pii/S0021999119308411.

Pakshal Bohra, Joaquim Campos, Harshit Gupta, Shayan Aziznejad, and Michael Unser. Learning activation functions in deep (spline) neural networks. *IEEE Open Journal of Signal Processing*, 1:295–309, 2020. doi: 10.1109/OJSP.2020.3039379.

Nicolas Boullé, Yuji Nakatsukasa, and Alex Townsend. Rational neural networks. In Hugo Larochelle, Marc'Aurelio Ranzato, Raia Hadsell, Maria-Florina Balcan, and Hsuan-Tien Lin (eds.), *Advances in Neural Information Processing Systems 33: Annual Conference on Neural Information Processing Systems 2020, NeurIPS 2020, December 6-12, 2020, virtual*, 2020. URL https://proceedings.neurips.cc/paper/2020/hash/a3f390d88e4c41f2747bfa2f1b5f87db-Abstract.html.

Djork-Arné Clevert, Thomas Unterthiner, and Sepp Hochreiter. Fast and accurate deep network learning by exponential linear units (elus). In Yoshua Bengio and Yann LeCun (eds.), *4th Inter-*

*national Conference on Learning Representations, ICLR 2016, San Juan, Puerto Rico, May 2-4, 2016, Conference Track Proceedings*, 2016. URL `http://arxiv.org/abs/1511.07289`.

Xavier Glorot, Antoine Bordes, and Yoshua Bengio. Deep sparse rectifier neural networks. In Geoffrey J. Gordon, David B. Dunson, and Miroslav Dudík (eds.), *Proceedings of the Fourteenth International Conference on Artificial Intelligence and Statistics, AISTATS 2011, Fort Lauderdale, USA, April 11-13, 2011*, volume 15 of *JMLR Proceedings*, pp. 315–323. JMLR.org, 2011. URL `http://proceedings.mlr.press/v15/glorot11a/glorot11a.pdf`.

Kaiming He, Xiangyu Zhang, Shaoqing Ren, and Jian Sun. Deep residual learning for image recognition. *arXiv preprint*, arXiv: 1512.03385, 2015a. URL `http://arxiv.org/arXiv:1512.03385`.

Kaiming He, Xiangyu Zhang, Shaoqing Ren, and Jian Sun. Delving deep into rectifiers: Surpassing human-level performance on imagenet classification. *arXiv preprint*, arXiv: 1502.01852, 2015b. URL `http://arxiv.org/abs/1502.01852`.

Dan Hendrycks and Kevin Gimpel. Bridging nonlinearities and stochastic regularizers with gaussian error linear units. *arXiv preprint*, arXiv: 1606.08415, 2016. URL `http://arxiv.org/abs/1606.08415`.

Diederik P. Kingma and Jimmy Ba. Adam: A method for stochastic optimization. In Yoshua Bengio and Yann LeCun (eds.), *3rd International Conference on Learning Representations, ICLR 2015, San Diego, CA, USA, May 7-9, 2015, Conference Track Proceedings*, 2015. URL `http://arxiv.org/abs/1412.6980`.

Günter Klambauer, Thomas Unterthiner, Andreas Mayr, and Sepp Hochreiter. Self-normalizing neural networks. In Isabelle Guyon, Ulrike von Luxburg, Samy Bengio, Hanna M. Wallach, Rob Fergus, S. V. N. Vishwanathan, and Roman Garnett (eds.), *Advances in Neural Information Processing Systems 30: Annual Conference on Neural Information Processing Systems 2017, December 4-9, 2017, Long Beach, CA, USA*, pp. 971–980, 2017. URL `https://proceedings.neurips.cc/paper/2017/hash/5d44ee6f2c3f71b73125876103c8f6c4-Abstract.html`.

Lu Lu, Yeonjong Shin, Yanhui Su, and George E. Karniadakis. Dying relu and initialization: Theory and numerical examples. *arXiv preprint*, arXiv: 1903.06733, 2019. URL `http://arxiv.org/abs/1903.06733`.

Alejandro Molina, Patrick Schramowski, and Kristian Kersting. Padé activation units: End-to-end learning of flexible activation functions in deep networks. In *International Conference on Learning Representations*, 2020. URL `https://openreview.net/forum?id=BJlBSkHtDS`.

Andrei Nicolae. PLU: the piecewise linear unit activation function. *arXiv preprint*, arXiv: 1809.09534, 2018. URL `http://arxiv.org/abs/1809.09534`.

Daniel W. Otter, Julian R. Medina, and Jugal K. Kalita. A survey of the usages of deep learning in natural language processing. *arXiv preprint*, arXiv: 1807.10854, 2018. URL `http://arxiv.org/abs/1807.10854`.

Adam Paszke, Sam Gross, Francisco Massa, Adam Lerer, James Bradbury, Gregory Chanan, Trevor Killeen, Zeming Lin, Natalia Gimelshein, Luca Antiga, Alban Desmaison, Andreas Kopf, Edward Yang, Zachary DeVito, Martin Raison, Alykhan Tejani, Sasank Chilamkurthy, Benoit Steiner, Lu Fang, Junjie Bai, and Soumith Chintala. Pytorch: An imperative style, high-performance deep learning library. In H. Wallach, H. Larochelle, A. Beygelzimer, F. d'Alché-Buc, E. Fox, and R. Garnett (eds.), *Advances in Neural Information Processing Systems*, volume 32. Curran Associates, Inc., 2019. URL `https://proceedings.neurips.cc/paper/2019/file/bdbca288fee7f92f2bfa9f7012727740-Paper.pdf`.

Prajit Ramachandran, Barret Zoph, and Quoc V. Le. Searching for activation functions. *CoRR*, abs/1710.05941, 2017. URL `http://arxiv.org/abs/1710.05941`.

Athanasios Voulodimos, Nikolaos Doulamis, Anastasios Doulamis, and Eftychios Protopapadakis. Deep learning for computer vision: A brief review. *Computational Intelligence and Neuroscience*, 2018:7068349, February 2018. ISSN 1687-5265. URL `https://doi.org/10.1155/2018/7068349`.

Yucong Zhou, Zezhou Zhu, and Zhao Zhong. Learning specialized activation functions with the piecewise linear unit. *CoRR*, abs/2104.03693, 2021. URL `https://arxiv.org/abs/2104.03693`.

