# OpenReview forum: "Neural networks with trainable matrix activation functions"
_ICLR.cc/2022/Conference — ICLR 2022 Submitted_

### Official Review · Reviewer_FB5F · 2021-10-28

**Correctness:** 2
**Technical Novelty And Significance:** 2
**Empirical Novelty And Significance:** 1
**Recommendation:** 1
**Confidence:** 4

**Main Review:**

I have several concerns with this paper, which I did not find well-written and significant enough for the following reasons.

- First, the introduction does not cover the literature on adaptive activation functions and lack several references. Hence the authors mention a few classical activation functions but omit to reference the adaptive ones proposed in the past few years such as swish (Ramachandran et al., 2017), rational functions (Molina et al., 2019; Boulle et al., 2020), splines (Bohra et al., 2020), adaptive standard activation functions for physics informed neural networks (Jagtap and Karniadakis, 2019), ...
Moreover, the authors write that it's hard to determine the optimal activation function for a specific application, while I agree about that, I want to point out the recent works about searching for activation functions as well as the whole literature on the approximation theory of neural networks that can provide some informations regarding the approximation power of neural networks with a specific activation functions. As an example, see the paper by Yarotsky (2017) giving bounds on the size of ReLU networks needed to approximate smooth functions. Papers proposing new activation functions usually try to provide theoretical justification or at least discuss some of these aspects but this is not the case here.

- The section 2, introducing the trainable matrix activation function is not written clearly. As an example, I do not understand the relation $\sigma \circ \eta(x)=D_l(\eta(x))\eta(x)$ if $D_l$ is a general diagonal function. I do not understand why we would need a matrix-vector multiplication (especially in the case where the matrix is diagonal), but also for non-diagonal matrix. There are already matrix-vector product before and after each activation functions so the units are already ``mixed'' in the network. Therefore, we can still assume that we need to apply an activation function on each individual unit. Then, to my understanding, the approach proposed by the authors is essentially equivalent to using trainable piecewise constant function with several breakpoints. This is basically a sort of disconnected splines so I do not see the main novelty with respect to the existing literature on this topic. Regardless of the novelty, one of my main concern about this approach is the resulting increase in the number of parameters in the neural networks.

- Later in this section, the authors consider nonlinear activation operators by using tri-diagonal matrices with trainable coefficients. I believe that there is a connection with the recent works trying to learn solution operators of partial differential equations, Green's functions, integral kernels,... (arXiv:2010.08895; arXiv:2003.03485; arXiv:2101.07206; arXiv:2105.00266; Lu et al, DeepONet, 2021). There should be a discussion about this in this section.

- The authors mention that the computational time is comparable to ReLU but do not report any timings to support this claim. Moreover, for the small network size that the authors employ in section 3, the increase of the network parameters will likely impact the training time.

- I do not find the experiments in section 3 significant enough to support the claims of the authors about the gain of the matrix activation functions over ReLU. In sections 3.1-3.2, the authors aim to approximation smooth functions and compare with ReLU. However, they use very shallow neural networks (1 or 2 hidden layers) and so the difference in accuracy is likely due to the increase in terms of trainable parameters. Moreover, it is known that deep neural networks have higher approximation power than shallow ones so I'm wondering why the authors did not test their approach on larger networks. The authors report training errors in their experiments but do not employ a testing set. In Fig.3(c), the constant training loss for ReLU seems to be due to the size of the network employed. The authors comment this by saying that "increasing the number of intervals in TMAF only lead to a few more training parameters", I disagree with this since the increase would grow as number of layers x number of neurons per layer so is very significant for the network size used. It is expected that a one/two hidden layer ReLU network won't be appropriate to approximate a sinusoid function as the neural network will basically be a piecewise linear function.

- Similarly, the experiments on CIFAR-10 and CIFAR-100 are not really conclusive as the authors employ a residual network instead of standard convolution networks. Moreover, they don't do any comparison with other (adaptive) activation functions. I also do not think that the reported accuracy is closed to the state-of-the-art. I encourage the authors to read the section 4.3 of (arXiv:1907.06732), which performs comparisons between several activation functions with different architectures on the CIFAR-10 dataset, The reported performances are all higher than 90%, which is way higher than the ones reported by the authors in Table 3.

Minor comments:
- in the abstract: optimizes weights, they are other typos in the manuscript
- no conclusions to summarize findings
- there is no code provided for reproducibility and assessment

**Summary Of The Paper:**

The authors introduce trainable matrix activation functions, consisting of trainable matrices at the activation layers to generalize ReLU, in order to increase the approximation performance of deep neural networks. Then, they evaluate their networks on approximation of functions and image classification examples, including the standard CIFAR-10 and CIFAR-100 datasets.

**Summary Of The Review:**

I find that the paper is not really well-written as it lacks references and the main contribution is not introduced clearly. Moreover, I think that the contributions are not significant enough and not well supported by experiments for ICLR. Hence, I recommend rejection of the paper.

---

> ### Author Response · Authors · 2021-11-22
> **Responses to Reviewer FB5F**
>
> Comment 1: First, the introduction does not cover the literature on adaptive activation functions and lack several references...
>
> Response 1: In the revised version, we added more references in the introduction.
>
>
> Comment 2: I do not understand why we would need a matrix-vector multiplication (especially in the case where the matrix is diagonal), but also for non-diagonal matrix.
>
> Response 2: We generalize ReLU to TMAF using matrix-vector multiplication, which is mathematically natural and simple. This generalization is different from classical generalization strategy such as piecewise constant or continuous continuous piecewise linear units in the literature.  Please have a look at the Remark on page 3.
>
>
> Comment 3: The authors mention that the computational time is comparable to ReLU but do not report any timings to support this claim. Moreover, for the small network size that the authors employ in section 3, the increase of the network parameters will likely impact the training time.
>
> Response 3: ReLU corresponds to a fixed TMAF with $m=2$ (two) intervals for each layer in the NN. In TMAF  we choose $m$ to be a fixed integer ($m=5$ is typical) and this asymptotically has the same computational cost as usual ReLU, namely,
>
> computational-cost(TMAF)  &lsim; ⁵⁄₂ (computational-cost(ReLU)
>
>
> Comment 4: The authors comment this by saying that "increasing the number of intervals in TMAF only lead to a few more training parameters", I disagree with this since the increase would grow as number of layers x number of neurons per layer so is very significant for the network size used.
>
> Response 4: In the revised version, we modify the notation. In fact, TMAF uses the same activation function for each neuron in the same layer. So the increase of cost would be only proportional to the number of layers and is independent of number of neurons.
>
> Comment 5: Similarly, the experiments on CIFAR-10 and CIFAR-100 are not really conclusive as the authors employ a residual network instead of standard convolution networks. Moreover, they don't do any comparison with other (adaptive) activation functions. I also do not think that the reported accuracy is closed to the state-of-the-art. I encourage the authors to read the section 4.3 of (arXiv:1907.06732), which performs comparisons between several activation functions with different architectures on the CIFAR-10 dataset, The reported performances are all higher than 90\%, which is way higher than the ones reported by the authors in Table 3.
>
> Response 5: We do appreciate this comment. State-of-the-art activation functions appear almost daily (weekly, monthly, quarterly). It is impossible to compare with all such new constructions. We have clearly shown that TMAF outperforms a representative set of NN in approximation. In our tests as representative set we use ReLU and Parametric ReLU (which includes Leaky ReLU). Overall the NN with TMAF has efficiency comparable to a ReLU NN. The accuracy of the TMAF-NN in approximating is better and is robust with respect to the oscillations in the data. It is reasonable to conclude that such a pool of numerical tests is sufficient to prove a concept and we expect to include in our future research a comparison with all competitive activation functions that are available in the literature.

---

> > ### Comment · Reviewer_FB5F · 2021-11-24
> > **Comment on the rebuttal**
> >
> > 1. I have read the revised introduction and it does not address my comment, the search for adaptive activation functions is an active research area with several contributions. The authors did not compare against any popular approach and their experiments have weaknesses, which makes it hard to support any claim of the form "Neural networks based on this approach are simple and efficient and are shown to be robust in numerical experiments." in the abstract.
> > 2. I read the remark but it does not contradict my previous comment: "Then, to my understanding, the approach proposed by the authors is essentially equivalent to using trainable piecewise constant function with several breakpoints. This is basically a sort of disconnected splines so I do not see the main novelty with respect to the existing literature on this topic." I also do not see the advantage of using discontinuous versus continuous piecewise linear activation functions in practice. If the authors see an increased accuracy, it is very likely due to the larger number of parameters in the activation function (to control the slope on both sides of the discontinuity), which in that case is not a fair comparison.
> > 3. I strongly disagree with this comment: since your activation function is trainable, you have to account for the extra gradient computations to also optimize the trainable parameters. A report of the computational time is needed to support any claim of this nature.
> > 4. See the point above.
> > 5. I did not ask to compare against all activation functions, but a subset of popular recent ones. Moreover, the setup of the experiments (section 3.1: shallow and high width networks while the theory tells us that the opposite should be preferred), (section 3.3: 6 years old neural network architectures) are not convincing at all.
> >
> > Based on these reasons, I decided to keep my score and recommendation.

---

### Official Review · Reviewer_MkDq · 2021-10-29

**Correctness:** 2
**Technical Novelty And Significance:** 2
**Empirical Novelty And Significance:** 1
**Recommendation:** 3
**Confidence:** 5

**Main Review:**

The paper has several major concerns that are listed below.

NOVELTY: the paper does not provide a review of related works and fails to mention connections to existing trainable (or untrainable) AFs. The form s(x)x is also the base of the Swish AF (where s(x) is the sigmoid) or its variants. Many AFs are built by piecewise linear or constant parts (s-shaped ReLU, adaptive piecewise linear unit, ...) (Apicella et al., 2021). The idea here is to combine Swish with a piecewise constant s(x), but the way it is described in the paper is convoluted (where the original AF is replaced with D(s(x))x, where D is a diagonal matrix).

The tri-diagonal AF is possibly novel but it is not tested properly, and it also fails to reference related works (e.g., the Maxout, or the bi-dimensional variants of kernel AFs).

COMPUTATIONAL COST: the paper says "it is observed that the computational time of D` and T` is comparable to the classical ReLU", but evaluting (6) or even worse (7) is clearly more expensive than a simple max(0, x). Concrete results on the overhead should be given to justify this sentence.

EXPERIMENTS: more realistic experiments should be provided, by comparing a larger set of trainable/fixed baselines (Swish, SELU, ...), letting the networks train to convergence (see, e.g., Fig. 7), provide standard deviation of the results, etc.

**Summary Of The Paper:**

The paper describes an approach to train activation functions. The main idea is to rewrite ReLU as s(x)x (where s(x) is either 0 or 1 depending on the value of x), and then generalize this with a piecewise constant s(x). The paper also proposes a generalization where values at different neurons can be combined. Finally, this is evaluated on two artificial problems and two standard computer vision benchmarks.

**Summary Of The Review:**

Some parts of the paper are novel, but the current exposition (and experimental results) are too shallow to warrant publication, which is why I am suggesting a rejection of the paper.

---

> ### Author Response · Authors · 2021-11-22
> **Responses to Reviewer MkDq**
>
> Comment 1: the paper does not provide a review of related works and fails to mention connections to existing trainable (or untrainable) AFs. The form s(x)x is also the base of the Swish AF (where s(x) is the sigmoid) or its variants. Many AFs are built by piecewise linear or constant parts (s-shaped ReLU, adaptive piecewise linear unit, ...) (Apicella et al., 2021). The idea here is to combine Swish with a piecewise constant s(x), but the way it is described in the paper is convoluted (where the original AF is replaced with D(s(x))x, where D is a diagonal matrix).
>
> Response 1: In the revised version, we have added more references in the introduction as well as in Remark on page 3.
>
>
> Comment 2: COMPUTATIONAL COST: the paper says "it is observed that the computational time of D and T is comparable to the classical ReLU", but evaluting (6) or even worse (7) is clearly more expensive than a simple max(0, x). Concrete results on the overhead should be given to justify this sentence.
>
> Response 2: We remove this sentence in the revised version. But for a finite number $m$ of intervals in the definition of TMAF (for example $m=2$ for ReLU), it is clear that the computational cost can be controlled and similar to that of ReLU or Parametric ReLU.
>
>
> Comment 3: EXPERIMENTS: more realistic experiments should be provided, by comparing a larger set of trainable/fixed baselines (Swish, SELU, ...), letting the networks train to convergence (see, e.g., Fig. 7), provide standard deviation of the results, etc.
>
> Response 3: In all experiments of the revised version, we compare TMAF with ReLU and/or the trainable Parametric ReLU.  These activation functions can provide general NN approximations with typical behavior close to the one exhibitted by the AF mentioned by the reviewer.

---

> > ### Comment · Reviewer_MkDq · 2021-11-24
> > **Answer to the rebuttal**
> >
> > Concerning Response 2, I am unconvinced computational cost is not an issue in the proposed approach, and this should be evaluated empirically in the paper. Concerning Response 3, most reviewers asked for an evaluation against state-of-the-art trainable AFs, of which PReLU is not a member. Because of these considerations, I prefer to keep my original score of the paper as the empirical evaluation is not on-par with the conference.

---

### Official Review · Reviewer_mR17 · 2021-10-31

**Correctness:** 1
**Technical Novelty And Significance:** 1
**Empirical Novelty And Significance:** 1
**Recommendation:** 3
**Confidence:** 3

**Main Review:**

The paper is very poorly written and as is is barely readable. The main ideas are not outlined with sufficient rigor and clarity. It is not clear what are the optimization parameters and how they are optimized. The empirical results are also not well described.

I don't understand the motivation in the beginning of Section 2:
- application of activation $\sigma$ returns a vector, while D returns a matrix. How can this be?
- $\eta_l$ is defined after it is used in the beginning of section 2.
- is $m$ a hyperparameter? How is it selected?

From what I understand the proposition in eq 6, this is just a series of multipliers to the linear activation of form $tu$, where $u$ is the pre-activated value and $t$ is the current slope. If this is true, then the final activation function won't even be continuous, right?

I also don't understand numerical results: how the params for $\alpha$ in Section 3.1 are selected? Are those trained? What about results in section 3.2? Why there are 100 intervals? Where does 2.01 came from? Generally, what is being trained? Are the results trained with eq 6 or eq 7? What are the final $t$ and $s$ values? Intervals presented define only the break points of $s$ values, what are the $t$ value?

Some other comments:
- Unfortunate re-use of the same variable that can be easily avoided: $f$ as a function and $f_n$ as the output.
- also, t in eq 1 and t in eq 6 mean different things.
- I would make the mathematical definition more rigor, e.g. a diagonal matrix-valued function mapping is not proper mathematical definition.
- entries from the discrete set {0, 1} = binary matrix
- the use of the compositional $\circ$ operator is very confusing and not-standard. It is quite hard to understand what operator is being applied and what is the resulting matrix dimension.


**Summary Of The Paper:**

The paper introduces a trainable matrix of activation functions. The authors propose to replace activation with a custom learnable piecewise linear function. The results are evaluated on a custom sine function with different oscillations as well as CIFAR-10 and CIFAR-100.


**Summary Of The Review:**

The paper is very poorly written and requires a significant improvements before it is ready for submission. As is, the paper is very far from begin ready even for a proper reviewer's evaluation.

---

> ### Author Response · Authors · 2021-11-22
> **Responses to Reviewer mR17**
>
> Comment 1: application of activation $\sigma$  returns a vector, while $D$ returns a matrix. How can this be?
>
> Response 1: $D$ returns a matrix. But in TMAF $D$ is multipplied with a column vector to obtain another column vector. Then, obviously, this operation results in an activation returning a vector. The multiplication by $D$ and a vector is not a dense vector matrix multiplication. If $D=\operatorname{diag}(z)$, then $D y= z\odot y$ where $\odot$ is the well known Schur product.
>
>
> Comment 2: $\eta_\ell$ is defined after it is used in the beginning of section 2.
>
> Response 2: In the revised version, we define $\eta_L=\eta_{L,\mathcal{W},\mathcal{B}}$ in Equation (2) to avoid such confusions.
>
>
> Comment 3: is $m$ a hyperparameter? How is it selected?
>
> Response 3: According to the defition of $\alpha_\ell,$ $m$ is the number of intervals used in $\alpha_\ell$. For ReLU we have $m=2$. The intervals are clearly given in each experiment and $m$ can vary in general. To keep the computational cost low, in our numerical experiments, we fix $m$ to be a small integer.
>
>
> Comment 4: I also don't understand numerical results:...
>
> Response 4: We have modified the notation in the revised version to clarify the results better.

---

> ### Comment · Area_Chair_sJm6 · 2021-11-26
> **Please respond to the author rebuttal**
>
> Dear Reviewer mR17,
> The authors have posted their rebuttal. I wonder whether the rebuttal addressed your concerns? Please respond to the authors. Thanks!
>
> AC

---

### Official Review · Reviewer_MSbu · 2021-11-01

**Correctness:** 3
**Technical Novelty And Significance:** 1
**Empirical Novelty And Significance:** 1
**Recommendation:** 1
**Confidence:** 4

**Main Review:**

My main concern with this paper is that there is already very similar work on this area of research, namely [1].
In [1], we find a learnable piecewise function which leads to the activation function PWLU. If we were to consider a different PWLU for every neuron in the network, we would have basically the same approach as the one introduced here.

The authors do not mention [1] as related work and of course, the novelty of the contribution is significantly constrained.

On the experimental section, there is no comparison to other learnable activation functions such as PAU, F-RELU or APL, PWLU, making it difficult for the reader to stack TMAF against the state of the art.
Moreover, the number of architectures and datasets evaluated is relatively small in comparison to other papers in the same area of research.
Finally, it seems like the paper only does one run for each of the experiments which might be problematic to estimate the actual performance of the activation function.

There are other minor improvements that could be done to the paper on the presentation front. It would be helpful for the reader to see the plots of the synthetic experiments overlapped with the ones produced by the networks. Although that space might be better used for a more comprehensive empirical evaluation.

[1] Zhou, Y., Zhu, Z., & Zhong, Z. (2021). Learning specialized activation functions with the Piecewise Linear Unit. arXiv preprint arXiv:2104.03693.


**Summary Of The Paper:**

The authors propose a new approach to activation functions in DNNs. More precisely, the authors present the trainable matrix activation function (TMAF), which is an activation realized by a matrix-vector multiplication whose entries are trainable.
The authors then present a diagonal and tri-diagonal operators. Here, these matrices represent a coefficient that multiplies the output of the layer of the neural network and thus can represent any general piecewise linear activation including relu and leaky relu.
These matrices are trainable and therefore add a small number of extra parameters to the network depending on the architecture of the layers.
Finally the authors demonstrate the benefits of TMAF in synthetic and real world datasets on a couple of architectures.

**Summary Of The Review:**

In short, there is very similar work in this area of research which was not mentioned by the authors.

---

> ### Author Response · Authors · 2021-11-22
> **Responses to Reviewer MSbu**
>
> Comment 1: My main concern with this paper is that there is already very similar work on this area of research, namely [1]. In [1], we find a learnable piecewise function which leads to the activation function PWLU. If we were to consider a different PWLU for every neuron in the network, we would have basically the same approach as the one introduced here.
>
> Response 1: We were not aware of the recent work on PWLU activation functions when writing the paper. In the revised version, we included remarks on the difference between PWLU and TMAF in the Remark on pages 2 and 3 and explanations under Equation (6). In particular, TMAF includes discontinuous activation functions as special cases while PWLU only allows continuous piecewise polynomials.
>
>
> Comment 2: On the experimental section, there is no comparison to other learnable activation functions such as PAU, F-RELU or APL, PWLU, making it difficult for the reader to stack TMAF against the state of the art. Moreover, the number of architectures and datasets evaluated is relatively small in comparison to other papers in the same area of research. Finally, it seems like the paper only does one run for each of the experiments which might be problematic to estimate the actual performance of the activation function.
>
> Response 2: We tested the approximation properties of the TMAF against a representative set of other learnable activation functions (parametric ReLU) and as our results convinsingly show that TMAF provides a competitive and robust approximation when compared to these NN approximations. A comprehensive comparisons with all of the NN constructions available would require substantial time and while it is one of the future research goals of our team we consider it falling beyond the scope of this paper.
>
>
> Comment 3: There are other minor improvements that could be done to the paper on the presentation front. It would be helpful for the reader to see the plots of the synthetic experiments overlapped with the ones produced by the networks. Although that space might be better used for a more comprehensive empirical evaluation.
>
> Response 3: The NNs (TMAF or other activation functions) fall in the category of nonlinear/adaptive approximation methods. The mathematical  theory shows that the nonlinear approximation results in errors which are always better than the ones provided by synthetic (linear) approximations. As this is well known fact, and because of the restricted space, we did not consider such comparison.

---

> > ### Comment · Reviewer_MSbu · 2021-11-28
> > **feedback**
> >
> > I appreciate the comments by the authors. However, I still disagree that a comparison to PReLU is enough for this kind of work at this conference.
> > I understand the computational challenges, but a stronger evaluation is still needed, including the time taken by the activation function.
> >
> > For those reasons, I would keep my score as is.

---

> ### Comment · Area_Chair_sJm6 · 2021-11-26
> **Please respond to the author rebuttal**
>
> Dear Reviewer MSbu,
> The authors have posted their rebuttal. I wonder whether the rebuttal addressed your concerns? Please respond to the authors. Thanks!
>
> AC

---

### Official Review · Reviewer_iGq4 · 2021-11-08

**Correctness:** 2
**Technical Novelty And Significance:** 2
**Empirical Novelty And Significance:** 1
**Recommendation:** 3
**Confidence:** 4

**Main Review:**

Strength:

The idea seems to be new.


Weakness:

1. The intuition why the proposed method is better is not provided. There are many ways to generalize the existing activation functions, but why would TMAF work?

2. Lack of thorough compassion to support the effectiveness of TMAF. There have been a lot of widely used activation functions, as mentioned in the introduction section: PLU, Leaky ReLU, PLU, Softplus, ELU, SELU, GELU etc. However, the authors did not compare TMAF with most of them, making the effectiveness of TMAF questionable.

3. The task of sin/cos function approximation is quite simple compared to fitting the highly non-convex neural networks. So section 3.1 and 3.2 might not be very informative.

4. The experiments on MNIST/CIFAR is based on very lousy baselines — The 2-hidden-layer network or ResNet18 is by no means the state-of-the-art methods, not even close. So using them in the experiments is not convincing. It would be better to have TMAF plugged in the latest networks, such as EfficientNet and Vision Transformers to test its efficacy.

5. The CIFAR-100 result is missing in Table 3, making the paper look incomplete and probably written in a hurry.

**Summary Of The Paper:**

This paper proposes a new type of activation function, called Trainable Matrix Activation Functions (TMAF), to replace the existing activation functions in neural networks, such as ReLU. TMAF is realized using matrix-vector multiplications, where entries of the matrices are trainable piecewise constants. Empirical studies show that TMAF can approximate the sin/cos type of functions better than ReLU/PReLU in terms of approximation error. TMAF also outperforms ReLU in MNIST/CIFAR classifications tasks.

**Summary Of The Review:**

All in all, the paper needs a lot more thorough experiments to justify the usefulness of the proposed activation functions.

---

> ### Author Response · Authors · 2021-11-22
> **Responses to Reviewer iGq4**
>
> Comment 1: The intuition why the proposed method is better is not provided. There are many ways to generalize the existing activation functions, but why would TMAF work?
>
> Response 1: We generalize ReLU to TMAF using matrix-vector multiplication, which is mathematically natural and simple. This generalization is different from classical generalization strategy in the literature. One advantage is that TMAF includes discontinuous activation functions and continuous as special cases. Please have a look at the Remark on page 3.
>
>
> Comment 2: Lack of thorough compassion to support the effectiveness of TMAF. There have been a lot of widely used activation functions, as mentioned in the introduction section: PLU, Leaky ReLU, PLU, Softplus, ELU, SELU, GELU etc. However, the authors did not compare TMAF with most of them, making the effectiveness of TMAF questionable.
>
> Response 2: In the revised version, we compared TMAF also with Parametric ReLU in the image classification problems. Parametric ReLU includes Leaky ReLU as a special case. TMAF outperforms Parametric ReLU on all examples that we have tested.
>
>
> Comment 3:
> The task of sin/cos function approximation is quite simple compared to fitting the highly non-convex neural networks. So section 3.1 and 3.2 might not be very informative.
>
> Response 3: We agree that section 3.1 is relatively simple. However, approximating a highly oscillating function, as the one in section 3.2, is a difficult task in general. As it is easily seen from the experiments in the manuscript, the standard network structures with gradient-descent-like training process do not work at all as approximants to such functions. In contrast, TMAF is able to produce good approximations.
>
>
> Comment 4: The experiments on MNIST/CIFAR is based on very lousy baselines — The 2-hidden-layer network or ResNet18 is by no means the state-of-the-art methods, not even close. So using them in the experiments is not convincing. It would be better to have TMAF plugged in the latest networks, such as EfficientNet and Vision Transformers to test its efficacy.
>
> Response 4: Our aim is to prove a concept, namely,  that TMAF outperforms the NN approximations using standard approaches. Although Resnet18 is not the state-of-the-art method, our numerical experiments provide evidence that TMAF is robust. As the referee rightfully mentions, employing TMAF in more efficient (the latest NN, using deeper layer structure) is desirable. It is, however, also a topic of our current and future research.

---

> > ### Comment · Reviewer_iGq4 · 2021-11-28
> > **I read the response, but will keep my score unchanged.**
> >
> > The response did not address my concerns well. In particular, the 'proof of concept' argument is not a legit reason to use a very weak baseline model nowadays. In fact, a lot of so-called novel techniques will not have advantages if the baseline is strong. So I will stick to my rating.

---

> ### Comment · Area_Chair_sJm6 · 2021-11-26
> **Please respond to the author rebuttal**
>
> Dear Reviewer iGq4,
> The authors have posted their rebuttal. I wonder whether the rebuttal addressed your concerns? Please respond to the authors. Thanks!
>
> AC

---

### Comment · Area_Chair_sJm6 · 2021-11-28
**Please post your post-rebuttal opinion!**

Dear Reviewers,

The authors have updated their manuscript and responded to your comments. Please check whether your concerns have been addressed and then post your further opinions *if you haven't*. This is the professional way to show respect to the authors' efforts. The deadline Nov. 29 is coming very soon. Thanks!

AC

---

### Decision · Program_Chairs · 2022-01-20

**Decision:**

Reject

**Comment:**

The paper proposed a new kind of activation function called matrix activation function that can be learnt jointly with the weights and biases. The paper got 2 strong rejects and 3 rejects. The major challenges include unclear motivation, limited novelty, incomplete related work, weak experiments, and poor paper writing. The author rebuttals did not convince the reviewers. The AC also read through the paper and agreed that the paper is below the bar of ICLR. In particular, the authors neglected a large literature of learning activation functions in the original version,

(two more examples:

[*] Xiaojie Jin, Chunyan Xu, Jiashi Feng, Yunchao Wei, Junjun Xiong, Shuicheng Yan: Deep Learning with S-Shaped Rectified Linear Activation Units. AAAI 2016: 1737-1743.

[#] Yan Yang, Jian Sun, Huibin Li, Zongben Xu: ADMM-Net: A Deep Learning Approach for Compressive Sensing MRI. NIPS 2017.
)

making them unable to compare with existing learnable activation functions thoroughly in the revised version in order to justify the necessity of using matrix activation functions. So the AC recommended rejection.